# Male ant reproductive investment in a seasonal wet tropical forest: Consequences of future climate change

David A. Donoso[1]*, Yves Basset[2,3,4,5⊙], Jonathan Z. Shik[6,7], Dale L. Forrister[8], Adriana Uquillas[9], Yasmín Salazar-Méndez[10], Stephany Arizala[7,11], Pamela Polanco[5], Saul Beckett[2], Diego Dominguez G.[1,12], Héctor Barrios[5⊙]

1 Departamento de Biología, Facultad de Ciencias, Escuela Politécnica Nacional, Quito, Ecuador, 2 ForestGEO, Smithsonian Tropical Research Institute, Balboa, Ancon, Panamá, 3 Faculty of Science, University of South Bohemia, Ceske Budejovice, Czech Republic, 4 Biology Centre of the Czech Academy of Sciences, Institute of Entomology, Ceske Budejovice, Czech Republic, 5 Maestría de Entomología, Universidad de Panamá, Panama City, Panamá, 6 Department of Biology, Centre for Social Evolution, University of Copenhagen, Copenhagen, Denmark, 7 Smithsonian Tropical Research Institute, Balboa, Ancon, Republic of Panama, 8 School of Biological Sciences, University of Utah, Salt Lake City, UT, United States of America, 9 Departamento de Matemáticas, Facultad de Ciencias, Escuela Politécnica Nacional, Quito, Ecuador, 10 Departamento de Economía Cuantitativa, Facultad de Ciencias, Escuela Politécnica Nacional, Quito, Ecuador, 11 Instituto de Biologia, Universidade Estadual de Campinas, (PG-IB/ UNICAMP), Campinas, SP, Brazil, 12 Departamento de Ciencias Naturales, Universidad Técnica Particular de Loja, Loja, Ecuador

⊙ These authors contributed equally to this work.
* david.donosov@gmail.com

**Data Availability Statement:** All relevant data are within the manuscript and its Supporting Information files.

**Funding:** DAD, AU and YZ were supported by EPN Proyecto de Investigación Grupal PIGR-19-16 to

## Abstract

Tropical forests sustain many ant species whose mating events often involve conspicuous flying swarms of winged gynes and males. The success of these reproductive flights depends on environmental variables and determines the maintenance of local ant diversity. However, we lack a strong understanding of the role of environmental variables in shaping the phenology of these flights. Using a combination of community-level analyses and a time-series model on male abundance, we studied male ant phenology in a seasonally wet lowland rainforest in the Panama Canal. The male flights of 161 ant species, sampled with 10 Malaise traps during 58 consecutive weeks (from August 2014 to September 2015), varied widely in number (mean = 9.8 weeks, median = 4, range = 1 to 58). Those species abundant enough for analysis (n = 97) flew mainly towards the end of the dry season and at the start of the rainy season. While litterfall, rain, temperature, and air humidity explained community composition, the time-series model estimators elucidated more complex patterns of reproductive investment across the entire year. For example, male abundance increased in weeks when maximum daily temperature increased and in wet weeks during the dry season. On the contrary, male abundance decreased in periods when rain receded (e.g., at the start of the dry season), in periods when rain fell daily (e.g., right after the beginning of the wet season), or when there was an increase in the short-term rate of litterfall (e.g., at the end of the dry season). Together, these results suggest that the BCI ant community is adapted to the dry/wet transition as the best timing of reproductive investment. We hypothesize that current climate change scenarios for tropical regions with higher average temperature, but

the ECOINT Research Group. This study was supported by SENACYT grant FID14-036 to HB and YB. Grants from the Smithsonian Institution Barcoding Opportunity FY013 and FY014 (to YB), and in-kind help from the Canadian Centre for DNA Barcoding allowed sequencing the insect specimens. JZS was funded by a European Research Council Starting Grant (ELEVATE: ERC-2017-STG-757810). Studies on insect population dynamics are supported by Czech Science foundation GAČR grant 20-31295S to YB. HB and YB are members of the Sistema Nacional de Investigación, SENACYT, Panama. The funders had no role in study design, data collection and analysis, decision to publish, or preparation of the manuscript. There was no additional external funding received for this study.

**Competing interests:** The authors have declared that no competing interests exist.

lower rainfall, may generate phenological mismatches between reproductive flights and the adequate conditions needed for a successful start of the colony.

## Introduction

The reproductive biology of ants has diversified across some 15,000 species [1–3]. In temperate areas, harsh winter conditions reduce the window for ant alates to mate and disperse. Most ant species mate in summer, after late spring snow but early enough for young colonies to raise the first workers and accumulate resources during fall [4, 5]. Instead, ant alates in tropical regions can potentially fly and mate across the entire year [3, 6, 7]. However, constant reproduction in time is the exception in tropical forests [7], and the extent to which ant species in tropical forests adjust their phenology to their surrounding environment is poorly understood [8]. This is unsettling as phenological mismatches between interacting species or between species and suitable environmental conditions to prosper are increasing due to global warming [9, 10].

The production of ant sexual forms and the start of a new colony are major steps for any ant species. Ant reproduction starts with the release of costly produced sexual alate individuals outside the nest [7]. Soon after mating, fertilized females must disperse and start a new colony. Males, instead, do not take part in the activities of young colonies and die [11]. Defenseless and solitary, the survival of mated females is fundamental for the local maintenance of a species. In these circumstances, the interplay of suitable nest sites, temperature, and water availability impact young colonies' survival in several ways [12, 13]. First, moist soils are ideal for queens digging a nest in the ground [14]. Second, increased water availability should decrease ant mortality due to desiccation. Third, water availability and elevated temperatures at the ecosystem level may increase litter decomposition by microbes and microarthropods (such as mites and collembolans), cascading into increased food availability for new ant colonies [15–17].

In tropical seasonal forests characterized by conspicuous dry and wet seasons [18], ant species could adjust their flight phenology to the environment in several ways [12, 13]. First, the timing of reproduction cycles should match the right environmental conditions. Ant alates usually fly towards the end of the dry season because new ant colonies, composed of gynes and the first workers, may benefit from environmental conditions often available in the wet season [12, 13]. Second, because male ants are ephemeral, successful mating of females and males originating from different colonies depends on flight synchronization in space and time [13, 19]. Flight synchronization depends on strong environmental signals for mature colonies to look after. Third, seasonality induces challenges to reproducing individuals with basic fitness constraints. For example, higher temperatures in the dry season may benefit energy-costly tasks such as flight, as the difference between ambient temperatures and the high temperatures needed by thoracic muscles of males to start movement is likely small [7, 20]. Finally, beyond water and temperature, ant communities depend on plant resources for food and space [16]. On Barro Colorado Island (BCI), a seasonal tropical rainforest in the Panama Canal, most plant productivity, in the form of branches and leaves, falls uneaten to the forest floor and accumulates during the dry season [15, 21]. Accumulating litterfall provides increased opportunities for female ants to locate suitable nest sites, all the more with a burst in productivity with the start of rainfall in May [17]. Thus, ant reproductive castes in seasonal forests may thus navigate within a thin line shaped by their physiological demands, limits imposed by sexual reproduction, and 'home' requirements of new colonies.

Despite recent advances in our understanding of ant male ecology, progress has been hindered either by the incomplete identification of species [8] or by a lack of modeling underlying abiotic factors affecting flight phenology. We performed a 1-yr intensive survey of male ants from Barro Colorado Island using Malaise traps. We overcame taxonomic issues by identifying the male ants with DNA COI barcodes. With this dataset, first, we described the reproductive phenology of male ants using circular statistics. Then, we aimed to answer the following questions: 1) Which environmental factors explain the intra-annual (weekly and seasonal) variability in male community composition? 2) What is the impact of environmental variables such as relative humidity, air temperature, rainfall, and litterfall on male ant abundance, a measure of reproductive investment?

## Materials and methods

### Study site

This study was conducted in Barro Colorado Island (BCI; [18]). BCI is a lowland seasonal wet forest located at 09˚09'19" N and 079˚50'15" W. BCI is a 1,500ha man-made island formed after 1914 when the Chagres River flooded its surrounding area [18]. Dry seasons usually extend from December to April and receive less than 300mm of the 2,600mm of annual rainfall. During the study period, the dry season started on November 22 (which was earlier than the average start date in mid-December) and ended, normally, on April 18. The annual average daily maximum air temperature is 26.3˚C. During the dry season, some tree species lose their leaves, increasing litterfall, and days that are usually 1˚C hotter due to direct sunlight [18]. Weather datasets are available at the Physical Monitoring Program of the Smithsonian Tropical Research Institute—http://biogeodb.stri.si.edu/physical_monitoring/research/barrocolorado. Weekly litterfall (g, dry mass of all fine litter; all leaves, flowers, fruits, seeds, small woody debris (<2 cm diameter) and "fine dust," summed over 62 litter traps, each consisting of 0.5-m$^2$ trapping area, for each census; [21]) was also included in our study as an explanatory variable.

### Male collection

We set 10 Malaise traps, model Townes [22], in the South-East corner of the 50 ha BCI Forest-GEO plot [23]. Traps were located at least 200 meters from each other [24]. They were checked weekly during 58 continuous weeks, from August 2, 2014 (traps set on July 26) until September 6, 2015 (traps set on August 31). Ants (Formicidae), sorted out from the Malaise samples, were highly biased towards males (97.75% of all specimens). Females, mainly attracted to light traps [25], were not considered here.

### Ethics statement

Collections of insects (including ants) in Barro Colorado Island are permitted under the general agreement of STRI and the government of Panama and do not require ethical committee approval as covered under the SMITHSONIAN DIRECTIVE 605 of August 11, 2014.

### Species delineation or identification

We identified males using morphological and molecular characters. First, male ants were sorted into species/morphospecies using standard morphological keys [26]. A subset of specimens (330 specimens from 136 species/morphospecies, out of 16,307 specimens and 161 ant species/morphospecies, see results below) was further identified using DNA barcodes, a piece of 658bp of the mitochondrial Cytochrome Oxidase I (COI) gene. These sequences were

compared against a more extensive database of 2,144 ant sequences from 371 ant species currently barcoded for the island. Laboratory work was done in collaboration with iBOL [27]. Sequences are deposited in the BOLD platform (www.boldsystems.org) [27] and freely available in the BOLD project BCIFO and BOLD dataset DS-BASSET12. We confirmed the ID of our specimens by estimating Neighbor-Joining trees using K2P distances (see [26] for further information on molecular techniques). BCI is home to nearly 400 ant species (D. Donoso, unpubl. data.), but most are not yet formally described. Thus, we used barcode identification numbers, or BINs, [27], as provisional identifications for those species currently without formal identification.

## Community-level analyses

First, we explored changes in the community-level composition of males along the study year by performing an NMDS with the metaMDS function in the R package Vegan [28]. We calculated pairwise Bray-Curtis dissimilarities using male incidence (i.e., a measure of ant abundance, obtained by summing the presence of ant males in the ten malaise traps in each week) of ant species between all weeks. Incidence values provide a measure of the ant community composition without bias on colony-level reproductive flight tactics, unknown for most ant species. We consider that stress values for the ordination of less than 0.2 indicate an appropriate NMDS solution. We fitted environmental variables [Rain at day 0 and accumulated rain from the last 7 and 14 days, Relative Humidity (0, 7 and 14), Maximum Temperature (0, 7 and 14), and litterfall (in g)] into the NMDS using the 'envfit' function in Vegan. We explored environmental variables' ability to explain NMDS axis 1 and 2 using Pearson correlations. Changes in species composition between dry and wet seasons were tested with analysis of similarities (ANOSIM, [29]) in Vegan. ANOSIM tests the null hypothesis that similarities in male assemblage composition are equal between seasons. ANOSIM provides a test statistic R, with values close to 1, meaning significant dissimilarity among groups. Monte-Carlo randomizations used season as group labels to test the hypothesis that within-group similarities were higher than expected by chance alone. The significance was assessed using a p-value of 0.05.

## Timing of males flying across the year

We used circular statistics to describe the timing of males flying at the species level [30]. Initially, we converted dates of male flying to angles from 0 to 360˚. Observations of male flight at the species level were weighted according to male abundance. We used male abundance because we considered the former a surrogate of ant reproductive investment. The mean angle, which represents the time of the year when most male flying occurs, was calculated [31]. Using Rayleigh's test, we tested whether the mean angle was nonrandom (i.e., equally dispersed in the year or aseasonal) using Rayleigh's test [32, 33]. A Rayleigh's test p-value above 0.05 indicates male flying is not concentrated on a particular date. These analyses were done in the R package *circular* [34]. We used a linear model to check if the time of the year when males are flying and subfamily can explain three measures of ant size (Wing Length, Wing Width, and Weber's Length). To perform this analysis, we re-scaled the mean angle for it to start not on January 1st but on November 22nd, when the dry season begins. Rescaling the mean angle implies that ants flying close to 0 degrees fly early in the dry season, and ants flying close to 360 degrees fly late in the wet season.

## Time-series model

Finally, we used a time series model to analyze how male abundance fluctuates across the sampled year. Like with the 'circular' analysis before, we used in the time-series model male

abundance because we were interested in ant reproductive investment. In our model, we seek to understand how the value of the dependent variable (the average male abundance per week, for the 58 weeks our survey lasted) varies when the value of one of the independent variables [daily air temperature (˚C), relative humidity (%), rainfall (mm), evapotranspiration (cm), litterfall (g)] changes.

Seeking to increase the explanatory power of the model, we built additional quantitative variables from original variables (S1 Table):

a.  For all variables, their averages, maximums, minimums, and the number of consecutive days showing increases (or decreases) during the previous 7 days of collection, the previous 2 and 3 weeks, the previous 1, 2, 3, and 6 months and the previous year of collection (e.g., the number of days within the last 3 weeks showing continuous decreases of rain; or the maximum amount of daily rain within the last 6 month).

b.  Accumulated values of all variables during the previous 7 days of collection, the previous 2 and 3 weeks, the previous 1, 2, 3, and 6 months and the previous year of collection. In addition, for each variable, the ratio between the accumulated values of the short and medium-term (e.g., accumulated 7 days of rain over accumulated 3 weeks of rain), and short and long-term (e.g., accumulated 2 weeks of rain over accumulated 1 year of rain).

c.  An indicator of the dry season (December to April) and wet season (May to November). This variable controlled for the seasonality of all other variables.

d.  Finally, the number of consecutive rainy days considering the previous 7 days before collection and the previous 2 and 4 weeks. For example, we scored as 3 any given week that showed 3 consecutive rainy days. Additionally, we looked for the interaction of this variable with the season (dry vs. wet), giving importance to rarer rain episodes in the dry season as possible flight triggers.

We estimated a Seasonal Autoregressive Integrated Moving Average with Exogenous Regressors (SARIMAX) model. We chose this model because it allowed us to set up a dynamic model of the relationship between the effects of independent variables on the dependent variable. Additionally, SARIMAX allowed us to control for seasonal patterns of the variables resulting from their dependence on the weather due to the nature of the variables analyzed, i.e., environmental variables that change according to the season (dry or rainy season). Ignoring the presence of seasonality in variables can increase the imprecision of the forecasts [35]. The general form of this model is written in Eqs (1) and (2):

$$Y_t = \beta_t X_t + u_t \tag{1}$$

and,

$$\phi_p(L)\Phi_P(L^s)\Delta^d\Delta_s^D u_t = A(t) + \theta_q(L)\Theta_Q(L^s)\varepsilon_t \tag{2}$$

where:

$\phi_p(L)$ is the non-seasonal autoregressive lag polynomial,

$\Phi_P(L^s)$ is the seasonal autoregressive lag polynomial,

$\Delta^d\Delta_s^D$ is the time series, differentiated $d$ times, and seasonally differentiated $D$ times,

$A(t)$ is the trend polynomial, including the intercept,

$\theta_q(L)$ is the non-seasonal moving average lag polynomial,

$\Theta_Q(L^s)$ is the seasonal moving average lag polynomial.

Eq (1) is just a linear regression of $X_t$ on $Y_t$, and Eq (2) describes the process followed by the error component as SARIMA (e.g., a simpler SARIMAX model that does not include the exogenous, or independent, variable). Initially, two model specifications were done to construct the SARIMAX model. While these two models did not show inconsistencies in the estimators obtained (i.e., the estimators did not show contradictory signs), we chose the best model using the Bayesian Information Criterion (BIC), which guaranteed that the preferred model followed the criterion of parsimony. The final specification and the postestimation analysis of the chosen model contained all variables explained below, including different combinations of weather variables. Then, to select final variables included in the model from these broader set of combinations (Table 1), carried out a Granger causality analysis [35]. This test determines if a variable "Granger causes" our dependent variable. If a variable X "Granger-causes" Y, then past values of X should contain information that helps predict Y above and beyond the information contained in past values of Y alone [35] (see S1 File). SARIMAX also requires variables to be stationary, meaning that the parameters of the variables do not change with time. To ensure this, we used the Augmented Dickey-Fuller test to identify the presence of unit roots in the stochastic component of the series [35], S1 File. Unit roots are present on non-stationary variables, and when found, variables must be differentiated to ensure stationarity.

## Results

Across the survey, we collected males of 16,307 ants representing 161 ant species/morphospecies. Twenty-one ant specimens in five species/morphospecies, in *Camponotus*, *Crematogaster*, *Hypoponera*, *Pheidole*, and *Pseudomyrmex*, were not identified with our barcode approach. *Rasopone arhuaca* (n = 58 weeks of occurrence), *Ectatomma ruidum* (n = 54), *Mayaponera constricta* (n = 52), *Strumigenys marginiventris* (n = 51), and *Strumigenys* AAP2320 (51) were the most common ants recorded in the Malaise traps flying continuously across the year. Three of these five continuous fliers were also the most abundant ants in our survey, *Rasopone arhuaca* (n = 7618 males), *Ectatomma ruidum* (n = 2610), and *Mayaponera constricta*

**Table 1. Description of variables included final SARIMAX model.**

| Variable | Description | Coeff. | Std. Error | t | p |
|---|---|---|---|---|---|
| **Accum_humidity_1y** | Accumulated humidity in the last year | -9.34 | 1.72 | -5.43 | <0.001 |
| **Min_temperature_7d** | Minimum temperature in the last week | 5.09 | 1.13 | 4.5 | <0.001 |
| **Accum_humidity_1y (3m lag)** | Accumulated humidity in the last year with a three-month lag | 8.03 | 1.26 | 6.38 | <0.001 |
| **Days_rain_7d (Dry Season)** | Consecutive days of rain in dry weather | 2.38 | 1.36 | 1.75 | 0.09 |
| **Decr_rain_2w** | Number of consecutive decreases in the amount of rain in the last 2 weeks | -8.26 | 1.74 | -4.76 | <0.001 |
| **Ratio_litterfall_7d/2w** | Ratio of accumulated litterfall during the previous week over accumulated litterfall during previous two weeks. This ratio is positive when short-term litterfall increases | -265.7 | 75.12 | 3.54 | 0.001 |
| **Average Abundance_7d** | Average abundance last week | 1.13 | 0.16 | 6.99 | <0.001 |
| **Average Abundance_2w** | Average abundance of last 2 weeks | -0.41 | 0.17 | -2.47 | 0.019 |
| **Average Abundance_7w** | Average abundance of last 7 weeks | 0.18 | 0.1 | 1.73 | 0.093 |
| **Model moving Average (5)** | Model moving average of order 5 | 0.93 | 0.03 | 29.78 | <0.001 |
| | **R-squared** | 0.88 | | Log likelihood | -158.8 |
| | **Adjusted R-squared** | 0.85 | | BIC | 8.65 |

(n = 589). A total of 64 species/morphospecies were collected as singletons or doubletons (S2 Table) and were removed from further analyses.

## Community-level analyses

The NMDS and ANOSIM analysis indicated that male ant composition differed between wet and dry seasons (R = 0.3; p = 0.001). Most environmental variables fitted into the NMDS ordination significantly correlated with community composition (Fig 1). Importantly, accumulated variables (e.g., accumulated relative humidity summed from the last 14 days) explained best community composition than environmental values obtained at the collection day (Table 2).

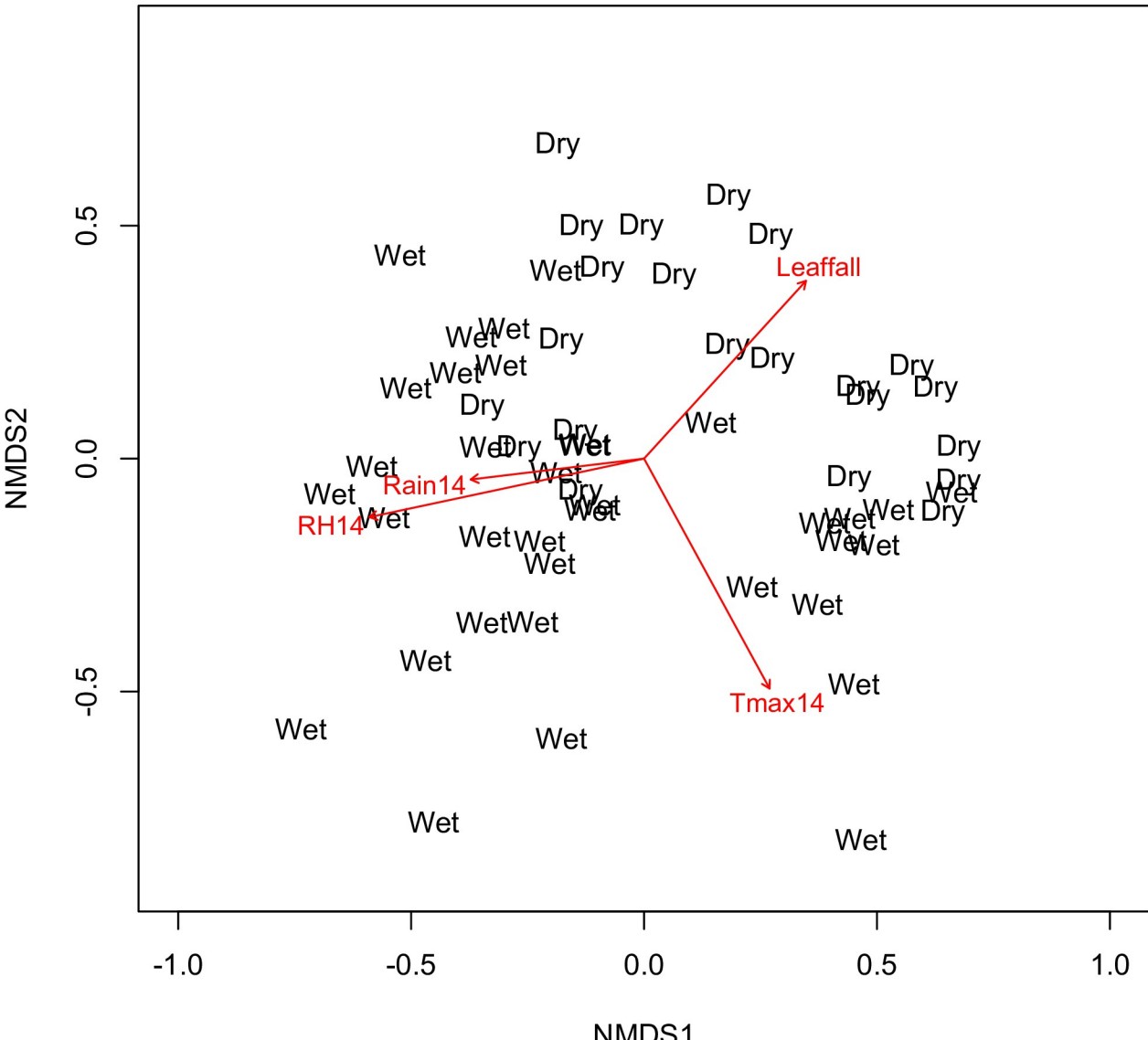

**Fig 1. A Non-Metric Multidimensional Scaling (NMDS) analysis of our male ant community across seasons.** The ordering of weekly samples in wet and dry seasons is based on Bray-Curtis distances. Environmental variables in Table 2 were fitted to the NMDS axes using Pearson correlations. For simplicity, only Rain, RH, Tmax (all accumulated from the last 14 days before collection), and litterfall are included in the figure. Stress = 0.14 for a three-dimensional ordination.

**Table 2. Fitted environmental variables into the NMDS axes.**

|  | NMDS1 | NMDS2 | $r^2$ | p |
|---|---|---|---|---|
| **Litterfall (0 days)** | 0.674 | -0.739 | 0.266 | >0.001 |
| **Rain (0 days)** | -0.647 | 0.763 | 0.033 | 0.406 |
| **Rain (7 days)** | -0.992 | 0.123 | 0.113 | 0.038 |
| **Rain (14 days)** | -0.993 | 0.120 | 0.141 | 0.011 |
| **RH (0 days)** | -0.998 | 0.058 | 0.114 | 0.037 |
| **RH (7 days)** | -0.971 | 0.238 | 0.297 | >0.001 |
| **RH (14 days)** | -0.978 | 0.208 | 0.363 | >0.001 |
| **Tmax (0 days)** | 0.519 | 0.855 | 0.192 | 0.004 |
| **Tmax (7 days)** | 0.584 | 0.812 | 0.336 | >0.001 |
| **Tmax (14 days)** | 0.479 | 0.878 | 0.316 | >0.001 |

Litterfall, Rain, Relative Humidity (RH), maximum daily Temperature at day 0, and accumulated values over 7 and 14 days from the survey were used for analysis.

### Timing of males flying across the year

While we found males flying throughout the year (Fig 2A), most ant species (81 out of 97) were seasonal, with male ants of most species peaking (mean angles) towards the end of the dry season (April) or at the start of the wet season (May) (Fig 2B, S3 Table). Only sixteen ant species were aseasonal (S3 Table). Furthermore, the time of the year when male flights of each species peak in time explains marginally wing width, wing length, and weber's length, and larger males with larger wings tended to occur later in the reproductive season (S5 Fig in S2 File).

### Time-series model

Male abundance remained at low and stable levels from mid-July to mid-December and grew gradually during the dry season from mid-December to the end of March. From March until

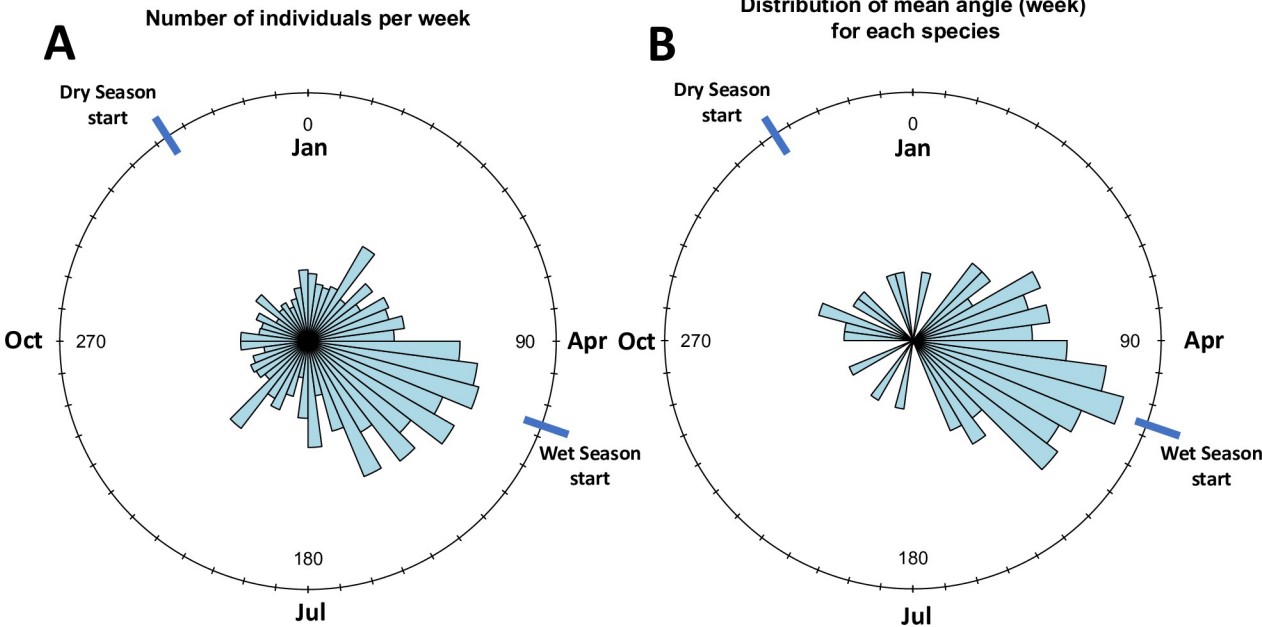

**Fig 2. Circular statistics results.** Average male abundance across the year (A) and mean angle of male flights for different ant species (B). Circles are divided into 48 bins, so each weak correspond broadly with one bin, and each month corresponds broadly with four bins.

mid-July, male abundance became higher in mean and more variable. The SARIMAX model found complex relationships between male abundance and environmental variables. For example, composite variables including relative humidity, temperature, rain, and litterfall explained male abundance (Table 1). As expected, controlling for the effects of all variables, the effects of accumulated humidity in the previous year to the week of collection (Accum_humidity_1y) and a decreasing level of rain during the last two weeks before collection (Decr_rain_2w) were negative, meaning that male abundance decreased during yearly peaks of humidity (e.g., during the wet season) but also in arid weeks. The ratio between the accumulated values of short- (seven days) and medium-term (two weeks) litterfall (Ratio_litterfall_7d_2w) also had a negative impact on ant abundance. On the other side, short-term increases in minimum temperature (Min_temperature_7d) and the number of consecutive days of rain during the dry season, measured in the previous seven days before collection, [Days_rain_7d (Dry Season)] showed a positive effect on male abundance. Finally, we found a significant positive effect of past male abundance (measured during the last seven days and seven weeks) on the male abundance of any given week, suggesting similar effects of environmental variables on the whole ant community.

## Discussion

Our analysis revealed that the phenology of the ant community was highly skewed towards the dry-wet season transition. While males of many ant species flew throughout the year, peaks of flying activity concentrated towards the end of the dry season and at the start of the rainy season. Moreover, we found wing size explained, in part, the time of flying, with larger males tending to flight later in the wet season. This variability suggests that environmental variables can ease specific demands of ant species to meet the needs of new colonies best. For example, species flying during the dry season may be under strong competitive selection for nesting sites, sorting themselves in time, and matching their flights with physiological requirements of fertilized queens to start a colony, with species flying earlier in the dry season being 1) less tolerant of desiccation or 2) with smaller wings, needing to accumulate less energy in reproductive tissue. On the other side, males of ant species flying on the wet season may instead successfully escape from constraints imposed by the difficulty of flying in the rain. Alternatively, colonies with larger ants may take more time to accumulate resources and build reproductive tissue and thus be forced to fly later in the year. We hypothesize that research focusing on female traits can go a long way in explaining these temporal differences in the composition of flying assemblages.

Male abundance, a measure of reproductive investment, of ant species in Barro Colorado Island peaked in a short window (March-May, peaking in April) within the year. This peak was depended on a balance of environmental variables (specific trends in relative humidity, rain, litterfall, and air temperature across the year). For example, male abundance decreased when 1-y accumulated humidity increased; in other words, males tended to fly at the time of the year when the ambient is driest. Because dry conditions are associated with low decomposition levels, which translates to lower food availability [16, 17], we hypothesize that water shortages prompt ant colonies to switch from growth to reproduction mode. However, male ant reproductive investment responded also to short-term variability in weather conditions. For example, increases in minimum temperature and short wet periods in the dry season elicit male flight. On the other hand, short-term dry periods (less than two weeks) restrain males from flying. Wet weeks in the dry season and dry weeks in the wet season are good predictors of seasonal changes. These relative 'slow-motion triggers' can be more reliable cues used by the ants to synchronize reproductive flights.

One worrying prediction of global warming is the potential mismatch between the phenologies of interacting species (e.g., flowers and their pollinators, [10, 36]), as temperature increases affect the timing of phenological events of interacting species differently. Weather patterns are changing in Panama, and the country is expected to be significantly wetter and warmer by the end of the century according to expected climate change trends in Panama [37, but see 38]. Since 1981, BCI has become more seasonal and has experienced an increase of 0.36˚C in mean annual temperature (mean temperature in January, in the dry season, has increased in 0.46˚C) and a 17.9% increase in mean annual precipitation (increases of rainy days = 13.3) [23]. Hotter days with no change (or decrease) in rain will increase water evaporation, increasing ambient dryness. Ants are thermophilic, and alate ants can use the extra heat to accelerate flight [7]. Thus, we hypothesize that ants flying in the dry season may initially benefit from increased temperature but ultimately could suffer phenological mismatches with plant-driven nest resources (i.e., litter), putting at risk the stability of the ant community in the island. Furthermore, extreme rains events are expected to increase as the world warms and may trigger earlier-than-expected reproductive flights [37]. These events are potentially harmful to new ant colonies, as queens fertilized earlier in the year would die of desiccation or from exhausting their fat resources before finding good nest sites.

Using DNA barcoding as an identification tool for a diverse seasonal tropical ant community, we show that male flights depend on interactions between environmental variables on short and medium time scales. Our understanding of alate ecology, which should include reproductive females, will benefit from studies exploring the limiting factors affecting early colony survival and establishment. Moreover, natural history observations of reproductive tactics incurred by the many ant species may help us underscore other causes of the encountered variability in this system [3]. While previous studies have focused on how climate change may impact ant communities through thermal performance traits of workers [39, 40], our results highlight complex performance bottlenecks linked to ant reproductive physiology and colony establishment. Ultimately, explaining how mating events determine ant community composition will require long-term datasets linking alate and worker abundance.

## Supporting information

**S1 Table. Environmental variables.**
(XLSX)

**S2 Table. Ant communities.**
(XLSX)

**S3 Table. Circular statistics results.**
(XLSX)

**S1 File. Analysis of stationarity and postestimation tests.**
(DOCX)

**S2 File. Relationship between male size and phenology variables.**
(DOCX)

## Acknowledgments

We thank ForestGEO and the Smithsonian Tropical Research Institute in Panama for logistical support. DAD thanks EPN for supporting travels to Panama and gives special thanks to F. Perez, Y. Lopez, R. Bobadilla, and A. Ramirez, for continuing support and logistics in Panama.

We thank for their time to two reviewers, which significantly improved the depth of ideas discussed in this manuscript.

## Author Contributions

**Conceptualization:** David A. Donoso, Yves Basset, Jonathan Z. Shik, Héctor Barrios.

**Data curation:** David A. Donoso, Yves Basset, Stephany Arizala, Pamela Polanco, Saul Beckett.

**Formal analysis:** David A. Donoso, Dale L. Forrister, Yasmín Salazar-Méndez, Diego Dominguez G.

**Funding acquisition:** David A. Donoso, Yves Basset, Jonathan Z. Shik, Héctor Barrios.

**Investigation:** David A. Donoso, Yves Basset, Dale L. Forrister, Yasmín Salazar-Méndez, Diego Dominguez G., Héctor Barrios.

**Methodology:** David A. Donoso, Yves Basset, Adriana Uquillas, Yasmín Salazar-Méndez, Stephany Arizala, Pamela Polanco, Saul Beckett, Héctor Barrios.

**Project administration:** Yves Basset, Héctor Barrios.

**Resources:** David A. Donoso, Yves Basset, Héctor Barrios.

**Supervision:** David A. Donoso, Yves Basset, Héctor Barrios.

**Validation:** David A. Donoso, Adriana Uquillas.

**Visualization:** David A. Donoso.

**Writing – original draft:** David A. Donoso.

**Writing – review & editing:** Yves Basset, Jonathan Z. Shik, Héctor Barrios.

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
