## [Decision Letter · Decision Letter 0]

7 Jan 2022

PONE-D-21-30447Male ant reproductive investment in a seasonal wet tropical forestPLOS ONE

Dear Dr. Donoso,

Thank you for submitting your manuscript to PLOS ONE. After careful consideration, we feel that it has merit but does not fully meet PLOS ONE’s publication criteria as it currently stands. Therefore, we invite you to submit a revised version of the manuscript that addresses the points raised during the review process.

Both reviewers have recommended accepting this manuscript for publication, pending a few minor changes. I agree with their assessment and am willing to consider a revised version for publication in the journal, assuming that you are able to modify the manuscript according to their recommendations.

We look forward to receiving your revised manuscript.

Kind regards,

Angelina Martínez-Yrízar, Ph.D.

Academic Editor

PLOS ONE

https://journals.plos.org/plosone/s/file?id=ba62/PLOSOne_formatting_sample_title_authors_affiliations.pdf”

“DAD, AU and YZ were supported by EPN Proyecto de Investigación Grupal PIGR-19-16 to the ECOINT Research Group. This study was supported by SENACYT grant FID14-036 to HB and YB. Grants from the Smithsonian Institution Barcoding Opportunity FY013 and FY014 (to YB), and in-kind help from the Canadian Centre for DNA Barcoding allowed sequencing the insect specimens. JZS was funded by a European Research Council Starting Grant (ELEVATE: ERC-2017-STG-757810). Studies on insect population dynamics are supported by Czech Science foundation GAČR grant 20-31295S to YB. HB and YB are members of the Sistema Nacional de Investigación, SENACYT, Panama. The funders had no role in study design, data collection and analysis, decision to publish, or preparation of the manuscript.”

“We thank ForestGEO and the Smithsonian Tropical Research Institute in Panama for logistical support. DAD, AU and YZ were supported by EPN Proyecto de Investigación Grupal PIGR-19-16 to the ECOINT Research Group. This study was supported by SENACYT grant FID14-036 to HB and YB. Grants from the Smithsonian Institution Barcoding Opportunity FY013 and FY014 (to YB), and in-kind help from the Canadian Centre for DNA Barcoding allowed sequencing the insect specimens. JZS was funded by a European Research Council Starting Grant (ELEVATE: ERC-2017-STG-757810). Studies on insect population dynamics are supported by Czech Science foundation GAČR grant 20-31295S to YB. HB and YB are members of the Sistema Nacional de Investigación, SENACYT, Panama.”

 “DAD, AU and YZ were supported by EPN Proyecto de Investigación Grupal PIGR-19-16 to the ECOINT Research Group. This study was supported by SENACYT grant FID14-036 to HB and YB. Grants from the Smithsonian Institution Barcoding Opportunity FY013 and FY014 (to YB), and in-kind help from the Canadian Centre for DNA Barcoding allowed sequencing the insect specimens. JZS was funded by a European Research Council Starting Grant (ELEVATE: ERC-2017-STG-757810). Studies on insect population dynamics are supported by Czech Science foundation GAČR grant 20-31295S to YB. HB and YB are members of the Sistema Nacional de Investigación, SENACYT, Panama. The funders had no role in study design, data collection and analysis, decision to publish, or preparation of the manuscript.”

Reviewers' comments:

Reviewer's Responses to Questions

**Comments to the Author**

1. Is the manuscript technically sound, and do the data support the conclusions?

Reviewer #1: Yes

Reviewer #2: Yes

2. Has the statistical analysis been performed appropriately and rigorously? 

Reviewer #1: Yes

Reviewer #2: Yes

3. Have the authors made all data underlying the findings in their manuscript fully available?

Reviewer #1: Yes

Reviewer #2: Yes

4. Is the manuscript presented in an intelligible fashion and written in standard English?

Reviewer #1: No

Reviewer #2: Yes

5. Review Comments to the Author

Reviewer #1: This is a very interesting study that draws on a very large data set that will be an important contribution to the literature on reproductive phenology in tropical ants. In interpreting the results, it would be helpful to consider the timescale for producing males, relative to the timescales of days and weeks in changing rainfall and temperature. How do species vary in the time needed to produce males? Are the responses to rainfall and humidity showing when males are produced or when they fly?

The manuscript should be edited for errors in English grammar that can be distracting and in a few places make the text difficult to understand.

Here are a few examples of issues that are just about translation, as well as one question about lines 104-106

line 46 and 334 'flew' not 'flied'

line 48 'dilucidated' - elucidated?

8 soon after mating

lines 104-106 - They fly in the dry season because resources are available in the wet season?

227 explanatory power of the model

207 ant size explains

334 throughout the year (not around the year)

342 tolerant of desiccation

345 by the difficulty of flying in the rain

Reviewer #2: The authors of this paper present and analyze valuable data in a vital area for which we have little quantitative understanding--the phenology of ant mating seasons. They do this through straightforward analyses that are clearly presented, and the entire paper flows smoothly. The findings are of interest to biologists interested in social insects, the evolution of phenology or life history, and strategies of reproductive investment. I list several comments below.

Introduction:

Lines 111-113: It would be worth elaborating how exactly higher temperatures might benefit them

Methods:

Line 145: Did the authors mean to say "increases" twice

Line 173: Out of how many specimens and species/morphospecies? This proportion can be calculated from the info in the Results but it would be useful to explicitly state the proportion here.

Lines 179-184: Did this method work to assign an ID or BIN to every male ant? Or were some unable to be matched to an existing species or BIN?

Lines 189-192: It is my understanding that Bray-Curtis dissimilarities are based on counts or abundance data, but the authors here say they used presence/absence matrices. How did the authors convert presence/absence data into abundances, or calculate Bray-Curtis dissimilarity for presence/absence data? Did they use the number of traps a species occurred in as abundance? The total number of males collected per week?

Line 220: Which one, abundance or incidence, did the authors use? This may relate to the comment above on lines 189-192. It appears from the following text that the authors used abundance.

Line 269: Unclear what the authors mean by SARIMA, as this term has not previously been introduced.

Lines 281-283/Appendix 2: It is unclear what the authors mean by stationary variables or unit roots.

Lines 309 and 337: The analysis appears to relate male size to time of year (angle, with 0 apparently being January 1), not time within the reproductive season.

Discussion:

Lines 337-339: It is unclear what the authors mean by "environmental variables are interacting... to meet the demands of new colonies"

Line 340: Under competitive selection for what resource? Nest sites? Mating times? It would help to clarify here.

Line 342: This seems to contradict lines 307-309 and Appendix 5, which found that larger males (those that are more desiccation resistant) flew later in the season, not earlier. If this statement is instead based on some other measure of desiccation resistance besides body size, it would be helpful if the authors explained it.

Lines 374-375: This sentence seems a little contradictory. The previous sentence said that BCI has been getting wetter, but this sentence describes the consequences of getting drier.

Lines 375-376: It is unclear what the authors mean by "use the extra heat to fuel flight." It would be helpful to elaborate.

Figures

Figure 1 caption: Change "scale" to "scaling"?

Figure 2/Figure 2 caption: The authors do not explain what the dark colored bar on the bottom right signifies

Appendix 5, Figure A5.1: The caption and the units imply that the y-axis show's Pearson's correlation coefficient, but the y-axis label says Mean Angle. It is also unclear what the line between points would signify, or why the graph in general is necessary. It seems to just be a list of 3 correlation coefficients (mean angle with 3 different measures of body size). If it is just a list of 3 values, the graph seems unnecessary and confusing. Also, the caption says the y-axis value for WingL should be 0.003, but the graph appears to show a value of 0.3.

6. PLOS authors have the option to publish the peer review history of their article (what does this mean?). If published, this will include your full peer review and any attached files.

Reviewer #1: No

Reviewer #2: No

---

## [Author Response · Author response to Decision Letter 0]

20 Feb 2022

We thank the reviewers for their time. We do think the comments improved a lot our manuscript. We hope that we addressed all comments well and that you find a more readable text. 

att

David A. Donoso, on behalf of all coauthors.

Review Comments to the Author

Reviewer #1: This is a very interesting study that draws on a very large data set that will be an important contribution to the literature on reproductive phenology in tropical ants. In interpreting the results, it would be helpful to consider the timescale for producing males, relative to the timescales of days and weeks in changing rainfall and temperature. How do species vary in the time needed to produce males? Are the responses to rainfall and humidity showing when males are produced or when they fly?

Response. Thank you very much for this comment. We agree this is a crucial point to make since male production (as opposed to male release or flight, the present study’s focus) is an important decision for an ant colony. Unfortunately, getting this data (i.e., when are males produced vs. males released?) is extremely difficult, and we only count with male release (flight) data. We have searched in the whole ms for sentences where we could have missed this point but could not find such sentences.

Also, it is essential to consider that in the time-series analyses, we were able to include composite variables that had temperature, humidity, and precipitation sums of large time scales (for example, Accum_humidity_1y (3m lag)) and that some of them explained flight data in a significant way. For instance, when Accumulated humidity in one year was highest, male flight decreased. We limited our Discussion to male flights and were reluctant to explain these results in terms of male production (because we do not have this data). Still, we hope future male production surveys will help understand the factor shaping male production.

The manuscript should be edited for errors in English grammar that can be distracting and in a few places make the text difficult to understand.

Response. Yes, the few native English speakers in our ms read and solve these grammar issues as much as possible, and we hope the ms is much clearer and cleaner. 

Here are a few examples of issues that are just about translation, as well as one question about lines 104-106

line 46 and 334 'flew' not 'flied'

Response. Done!

line 48 'dilucidated' - elucidated?

Response. Done!

8 soon after mating

Response. Done!

lines 104-106 - They fly in the dry season because resources are available in the wet season?

Response. YES! In this sentence, we compare male flights (dry season) vs colony demands (wet season). To improve the readability of this sentence, we replace it with this new one “Ant alates usually fly towards the end of the dry season because new ant colonies, composed of gynes and the first workers, may benefit from environmental conditions that are often available in the wet season”

227 explanatory power of the model

Response. Done!

207 ant size explains

Response. Done!

334 throughout the year (not around the year)

Response. Done!

342 tolerant of desiccation

Response. Done!

345 by the difficulty of flying in the rain

Response. Done!

Reviewer #2: The authors of this paper present and analyze valuable data in a vital area for which we have little quantitative understanding--the phenology of ant mating seasons. They do this through straightforward analyses that are clearly presented, and the entire paper flows smoothly. The findings are of interest to biologists interested in social insects, the evolution of phenology or life history, and strategies of reproductive investment. I list several comments below.

Introduction:

Lines 111-113: It would be worth elaborating how exactly higher temperatures might benefit them

Response. We rewrote the sentence in the following way. “For example, higher temperatures in the dry season may benefit energy-costly tasks such as flight, as there is a small difference between ambient temperatures and the high temperatures needed by thoracic muscles of males to start movement”

Methods:

Line 145: Did the authors mean to say "increases" twice

Response. Sorry for this. Fixed! The second ‘increases’ is now deleted.

Line 173: Out of how many specimens and species/morphospecies? This proportion can be calculated from the info in the Results but it would be useful to explicitly state the proportion here.

Response. Done! We added the totals. The new sentence now reads “A subset of specimens (330 specimens from 136 species/morphospecies, out of 16,307 specimens and 161 ant species/morphospecies, see results below)”

Lines 179-184: Did this method work to assign an ID or BIN to every male ant? Or were some unable to be matched to an existing species or BIN?

Response. Good point. We were so happy with the barcode approach, we forgot to write down that 21 ant specimens in 5 species/morphospecies were not able to be identified (mostly because we could not barcode every single specimen). We have included this info in the Results in the new version of the ms.

Lines 189-192: It is my understanding that Bray-Curtis dissimilarities are based on counts or abundance data, but the authors here say they used presence/absence matrices. How did the authors convert presence/absence data into abundances, or calculate Bray-Curtis dissimilarity for presence/absence data? Did they use the number of traps a species occurred in as abundance? The total number of males collected per week?

Response. You are right. Bray-Curtis must be done in Abundance data. We indeed did our NMDS on Incidence, which is a measure of abundance data and, as you mentioned, is the summed number of presences in all traps in each week. We have amended the text in the new version.

Line 220: Which one, abundance or incidence, did the authors use? This may relate to the comment above on lines 189-192. It appears from the following text that the authors used abundance.

Response. We used abundance (not incidence). We have corrected the text. Sorry for the typos.

Line 269: Unclear what the authors mean by SARIMA, as this term has not previously been introduced.

Response. Thanks for calling our attention to this mistake. We have now included in the text that a SARIMA is a “simpler SARIMAX model that does not include the exogenous, or independent, variable”.

Lines 281-283/Appendix 2: It is unclear what the authors mean by stationary variables or unit roots.

SARIMAX also requires variables to be stationary, meaning that the parameters of the variables do not change with time. To ensure this, we used the Augmented Dickey-Fuller test to identify the presence of unit roots in the stochastic component of the series [36], Appendix 2 [36]. Unit roots are present on non-stationary variables, and when found, variables must be differentiated to ensure stationarity.

Lines 309 and 337: The analysis appears to relate male size to time of year (angle, with 0 apparently being January 1), not time within the reproductive season.

Response. Thanks for catching this up. You are right. We forgot to mention, and we have added to the main text that… “to perform this analysis, we re-scaled the mean angle for it to start not on January 1st, but on November 22th, when the dry season starts, rescaling the mean angle imply that ants flying close to 0 degrees fly early in the dry season, and ants flying close to 360 degrees fly late in the wet season” 

Discussion:

Lines 337-339: It is unclear what the authors mean by "environmental variables are interacting... to meet the demands of new colonies"

Response. We have rewritten this sentence and now reads. “This variability suggests that environmental variables can ease specific demands of ant species to best meet the needs of new colonies”

Line 340: Under competitive selection for what resource? Nest sites? Mating times? It would help to clarify here.

Response. For mating sites. We have added this in the main text of the new version.

Line 342: This seems to contradict lines 307-309 and Appendix 5, which found that larger males (those that are more desiccation resistant) flew later in the season, not earlier. If this statement is instead based on some other measure of desiccation resistance besides body size, it would be helpful if the authors explained it.

Response. Sorry. Another mistake. We meant less tolerant of desiccation! Amended!

Lines 374-375: This sentence seems a little contradictory. The previous sentence said that BCI has been getting wetter, but this sentence describes the consequences of getting drier.

Response. Yes, as we mention in the text, two modeling studies from the physical (not biological) literature disagree with projections of weather patterns in the Panama Canal area for the future, with Fabrega et al. 2013 predicting a dry future and Imbach et al. 2018 predicting a wet future. To break the tie, we decided to follow and interpret our results in the light of a third study (Anderson-Texeira 2015) who provides strong evidence that BCI has become, in the last 30 years, hotter (especially in the dry season), and wetter (with a large increase of rainy days in the wet season). We have modified our text for these patterns, and our interpretations of weather changes’ effects on male abundance, to become clearer.

Lines 375-376: It is unclear what the authors mean by "use the extra heat to fuel flight." It would be helpful to elaborate.

Response. Please see the previous answer on this same topic. We reworded the sentence in this way: “Ants are thermophilic, and alate ants can use the extra heat to accelerate flight [7].

Figures

Figure 1 caption: Change "scale" to "scaling"?

Response. Error fixed.

Figure 2/Figure 2 caption: The authors do not explain what the dark colored bar on the bottom right signifies

Response. Sorry. A stupid mistake on our part. The bar has been removed.

Appendix 5, Figure A5.1: The caption and the units imply that the y-axis show's Pearson's correlation coefficient, but the y-axis label says Mean Angle. It is also unclear what the line between points would signify, or why the graph in general is necessary. It seems to just be a list of 3 correlation coefficients (mean angle with 3 different measures of body size). If it is just a list of 3 values, the graph seems unnecessary and confusing. Also, the caption says the y-axis value for WingL should be 0.003, but the graph appears to show a value of 0.3.

Response. As suggested, we have removed the results of the correlation (that were indeed badly retyped into our text), and now we focus on the regression results.

---

## [Editor Report · Decision Letter 1]

17 Mar 2022

Male ant reproductive investment in a seasonal wet tropical forest: consequences of future climate change

PONE-D-21-30447R1

Dear Dr. Donoso,

We’re pleased to inform you that your manuscript has been judged scientifically suitable for publication and will be formally accepted for publication once it meets all outstanding technical requirements.

Kind regards,

Angelina Martínez-Yrízar, Ph.D.

Academic Editor

PLOS ONE

Additional Editor Comments (optional):

Please revise proofs carefully to correct several minor errors, such as missing or extra spaces between words or sentences, or misplaced punctuation.
---

## [Editor Report · Acceptance letter]

23 Mar 2022

PONE-D-21-30447R1 

Male ant reproductive investment in a seasonal wet tropical forest: consequences of future climate change 

Dear Dr. Donoso:

I'm pleased to inform you that your manuscript has been deemed suitable for publication in PLOS ONE. Congratulations! Your manuscript is now with our production department. 

Kind regards, 

on behalf of

Dr. Angelina Martínez-Yrízar 

Academic Editor

PLOS ONE